# The Antioxidant Effect of Burdock Extract on the Oxidative Stability of Lard and Goose Fat during Heat Treatment

**DOI:** 10.3390/foods13020304

**Published:** 2024-01-18

**Authors:** Flavia Pop, Thomas Dippong

**Affiliations:** Department of Chemistry and Biology, Faculty of Science, Technical University of Cluj-Napoca, 76A Victoriei St., 430122 Baia Mare, Romania; flavia_maries@yahoo.com

**Keywords:** goose fat, lard, burdock extract, heating, oxidative stability

## Abstract

Concerns regarding product quality and nutrition are raised due to the effects of high temperatures on frying fats. The aim of this research was to examine the effects of temperature and burdock extract addition in relation to quality parameters for dietary lard and goose fat exposed to heating. In order to monitor quality changes, animal fats and 0.01% additivated fats were heated at different temperatures (110, 130, 150, 170, 190, and 210 °C for 30 min). Thiobarbituric acid-reactive substances test (TBARS), peroxide value (PV), iodine value (IV), acid value (AV), saponification value (SV), total polar compounds (TPoC), total phenolic content (TPC), fatty acid (FA) content, and microscopic examination were established in order to quantify the level of oxidative rancidity. Heating temperature and additivation had a significant (*p* < 0.001) effect on peroxide value. In all fats, values of thiobarbituric acid-reactive substances significantly (*p* < 0.001) increased with heating temperature, but values decreased when burdock extract was added in a proportion of 0.01%. Positive correlations were found between AV and PV for lard (*r* = 0.98; *p* < 0.001) and goose fat (*r* = 0.96; *p* < 0.001). The heating temperature had a significant effect on total MUFAs in both lard and goose fat (mostly in non-additivated fat). Statistical analysis of the data showed that the addition of burdock extract at a concentration of 0.01% significantly (*p* < 0.01) reduced the installation of oxidation process in alimentary fats heated at different temperatures. Animal fats were well protected from oxidation by burdock extract, which demonstrated its efficacy as an antioxidant; it may be used to monitor the fats oxidation and to estimate their shelf-life stability.

## 1. Introduction

Since it allows for quick product preparation, frying is a widely utilized food preparation technique. The impact of high temperatures on fats and oils subjected to frying is a serious concern for both product quality and nutrition because of the widespread use of these fats. Fried foods are very popular among consumers because they have desirable sensory properties such as crunchy texture, appealing colour, and unique flavours [1]. However, fats and oils undergo a variety of chemical reactions at high frying temperatures which change the fried product’s flavour and the frying fat’s quality. The quality of frying fat has a major impact on the taste, flavour, consumer acceptance, and shelf life of the fried food [2].

Obtaining alimentary animal fats consists of melting fats obtained from raw materials at a temperature between 65–75 °C, and then filtering them to remove water and solid contaminants. A significant amount of these fats are used in Romanian cooking and frying of various foods. Using animal fats in food preparation has nutritional benefits. The fat-soluble vitamins A, E, D, and K, as well as important fatty acids, are transported by dietary animal fats. Additionally, fats help these vitamins to be absorbed and transport the vitamins and their precursors throughout the body. Dietary fats improve the flavour, texture, and smell of the food. Moreover, they take more time to be digested in the stomach than proteins or carbohydrates, delaying the feeling of hunger. Fats not only supply energy but also serve as the precursors of glycolipids and phospholipids, two crucial components in cell membranes [3]. Several research studies have shown a correlation between the consumption of animal fats in a balanced diet and a lower incidence of cardiovascular illnesses and related risk factors, including obesity, insulin resistance, and tumours [4,5,6].

Common chemical processes when frying fat include hydrolysis, oxidation, and polymerization, which result in the production of volatile or nonvolatile chemicals. These reactions speed up the decomposition of the used frying fat by altering and modifying its chemical composition [7]. Glycerides undergo partial hydrolysis, unsaturated fatty acids undergo oxidative degradation, and the glycerol becomes dehydrated and transforms into acrylic aldehyde throughout the heat process. Changes are encouraged by the faster rate of oxidation and reactivity of the hydrolysis products, free fatty acids, mono- and diglycerides, compared to the original triacylglycerides [8,9]. Fats undergo oxidation, which results in the production of aldehydes, ketones, and dimerized triglycerides through oxygen bridges, free or oxidized fatty acids, etc. Factors that increase oxidation include temperature, the type of fat, the fat/air contact, and the rate at which food absorbs fat [10]. The primary reaction resulting from high heat treatment is the cyclization of fatty acid molecules. The dimers and polymers of triacylglycerides are the most significant class of alteration products from a quantitative perspective, and the high temperatures attained throughout the process also catalyse their synthesis [11,12]. New alterations occur during the autoxidation process, including physical (an increase in viscosity and scum production), chemical (creation of polymers, volatile chemicals), and organoleptic (flavour alteration, palatability, darkening). The fried food absorbs the non-volatile substances which are kept in the fat and end up in the consumer’s diet [13].

Since animal fats are primarily saturated and monounsaturated fats, they are more heat-resistant and have a longer shelf life than vegetable fats. Reduced oxidation in animal fats makes them less vulnerable to the toxins and cancer-causing substances produced when using vegetable oil. Owing to their greater stability, food cooked in animal fats absorbs less oil and fat [14]. Foods and vegetable oils which are rich in unsaturated fatty acids are more vulnerable to oxidation. The oil is increasingly prone to oxidation and thermal degradation as more highly unsaturated double bonds are present in the sample. Compared to unsaturated oils, animal fats are safer and produce less carcinogenic compounds [9].

In the food industry, antioxidants are frequently utilized, especially those derived from plants. Extracts made from the various burdock plant components have a range of pharmacological and biological properties, including antioxidant, anticancer, anti-inflammatory, antidiabetic, antiviral, and antimicrobial effects. It was stated that burdock extract contains chlorogenic, quinic, caffeic, *p*-coumaric, cinnamic, and gallic acids. Additionally, the functional qualities of burdock inulin were demonstrated, as it showed a better oil-holding capacity and promising swelling characteristics [15].

The antioxidant activity of phenolic compounds such us gallic, gentisic, protocatechuic, syringic, vanillic, caffeic, and ferulic acids was studied in pork lard [16,17,18]. It has been established that natural antioxidants are effective in stabilizing lard’s oxidative process and the antioxidant activity declined with increasing temperatures. The effect of green tea extract on quality parameters and shelf life of animal fats during storage was investigated [19]. Organoleptic analyses revealed that, after a month of storage, no deterioration was seen in samples with green tea extract. The benefits of adding plant extracts such as rosemary extract, oregano, black tea, sage, and thyme to lard in order to inhibit lipid oxidation have been the subject of various reports [20,21,22]. Their activity was effective in the inhibition of fats oxidation at a concentration of 0.01%, and also exhibited antimicrobial efficiency.

Investigations assessing the antioxidative properties of natural antioxidants in dietary fats are still lacking, and to the best of our knowledge, no investigations have been conducted concerning the effects of burdock extract on the oxidative stability of lard and goose fat subjected to heating.

The goal of this study was to examine the impact of varying temperatures (110, 130, 150, 170, 190, and 210 °C) and the addition of burdock extract at a concentration of 0.01% on the qualitative characteristics of lard and goose fat exposed to heating. Iodine value (IV), peroxide value (PV), thiobarbituric acid-reactive substances (TBARS), acid value (AV), saponification value (SV), total polar compounds (TPoC), total phenolic content (TPC), fatty acid (FA) content, and microscopic examination are among the methods utilized for the assessment of fats stability and monitoring of their degradation when heating. The oxidation process in lard and goose fat subjected to heating was reduced by the addition of burdock extract in a proportion of 0.01%.

## 2. Materials and Methods

### 2.1. Materials

Two categories of dietary fats were the subject of the study: lard and goose fat. These fats were selected based on the differences and similarities between them. They differ in the composition of polyunsaturated fatty acids (goose fat is richer compared to lard), but they have a similar colour (white-yellowish) and similar consistency, and both can be used as spreads or frying fats. Raw materials were collected immediately after slaughtering. Mangalica pig fat from the recently obtained subcutaneous adipose tissue was melted at 75 °C on a water bath. Goose fat was extracted from male and female goslings of the Landaise breed at 18 weeks of age. Animals’ diets were based on soybean and corn meal, with phosphorus, salt calcium, vitamins, and minerals. Animals’ diets were not enhanced with long-chain polyunsaturated fatty acids or antioxidants. Approximately 500 g of raw fat was divided into small pieces, heated at 75 °C on a water bath, centrifuged, and filtered. The burdock was purchased from the local market. The leaves were manually separated, cleaned with active chlorine, then dried in an air oven at 50 °C to a constant weight. The solvent extraction method was used to obtain the extract. Crushed leaves were dipped into a 70:29.5:0.5 *v*/*v*/*v* mixture of acetone, ethyl alcohol, and acetic acid, with a 1:10 ratio for dehydrated leaves and solvent. After the mixture was triturated in a homogenizer equipment, Velp OV 5 model (Velp Scientifica, Usmate Velate, Italy), it was vacuum filtered through a grade 2 paper filter. The supernatants were then mixed. A rotary evaporator was utilised to evaporate the remaining solvent at 60 °C [23]. 

In the melted fats, burdock extract was dissolved in the proportion of 0.01% (0.01 g/100 g of fat). One sample was used without heating for each fat under study. About 50 g of fat was heated in an electric oven HBG633NB1 model (Bosch GmbH, Gerlingen-Schillerhöhe, Germany) at 110, 130, 150, 170, 190 and 210 °C for 30 min. Glass vessels with an air-exposed surface area of 80 cm^3^ were used for each fat and heating temperature. Before being analysed, the cold samples were placed in glass tubes and refrigerated. Every sample was examined using three replications. All chemicals utilised were of analytical quality and were procured from Merck (Merck, Darmstadt, Germany).

### 2.2. Methods

#### 2.2.1. Peroxide Value (PV) Determination

The UV-VIS T60U spectrophotometer (Bibby Scientific, London, UK) was used to measure the peroxide value. Its operating temperature range was 5–45 °C, its field wavelength was 190–1100 nm, and its wavelength precision was 0.1 nm. The procedure was established on the spectrophotometer measurement of ferric ions (Fe^3+^), which are produced when hydroperoxides oxidize ferrous ions (Fe^2+^) in the presence of ammonium thiocyanate (NH_4_SCN). Each solution’s absorbance was measured at 500 nm. Thiocyanate ions (SCN^−^) combine with Fe^3+^ ions to create a red-violet homogeny that can be measured spectrophotometrically. PV was measured by creating a calibration curve (absorbance at 500 nm vs. Fe^3+^, reported in µg). The peroxide value was given as meq O_2_ kg^−1^ fat [24].

#### 2.2.2. Thiobarbituric Acid-Reactive Substances Test (TBARS)

The following procedure was used to determine TBARS: TBA Reagent (0.02 M 2-thiobarbituric acid in 90% glacial acetic acid) was obtained, then a glass-stoppered test tube containing 1 g of fat sample was filled with 5 mL of TBA reagent. The contents of the tube were combined after it was stopped. The tube was then immersed for 35 min in a boiling water bath. A blank for the TBA reagent in distilled water was made and handled the same as the sample. The sample was heated and then allowed to cool for ten minutes in tap water. After transferring a portion to a cuvette, the sample’s optical density was read against the blank at a wavelength of 538 nm in a T60U spectrophotometer (Bibby Scientific, London, UK) model spectrophotometer. By creating suitable dilutions of the 1 × 10^−3^ M 1,1,3,3-tetraethoxypropane standard solution, a standard curve was created to provide amounts that vary from 1 × 10^−8^ to 7 × 10^−8^ mol of malondialdehyde in 1 mL. The optical densities of these dilutions were measured at a wavelength of 538 nm after they were treated with TBA reagent. The TBARS value was stated as mg malondialdehyde (MDA) kg^−1^ fat [25].

#### 2.2.3. Iodine Value (IV) Determination

The Hanus method was used to determine the iodine value. To halogenate the double bonds, 0.5 g of sample (dissolved in 15 mL CCl_4_) was combined with 25 mL of Hanus solution (IBr). After 30 min in the dark, the mixture was exposed to 20 mL of KI (100 g/L) and 100 mL of distilled water, which converted the excess IBr to free I_2_. Titration was used to quantify free I_2_ with 24.9 g/L Na_2_S_2_O_3_·5H_2_O using starch (1.0 g/100 mL) as an indicator. The iodine value was represented as g I_2_ 100 g^−1^ fat [24].

#### 2.2.4. Acid Value (AV) Determination

Acid value (AV) was determined by using phenolphthalein as an indicator and sodium hydroxide solution 0.1 N to neutralize the sample’s acidity. A conical flask containing 2 g of the material was filled with 50 mL of 99% ethanol that had been neutralized with 0.1 N NaOH and phenolphthalein as an indicator. After shaking the flask, two drops of the phenolphthalein indicator solution were added, then the mixture was neutralized by adding 0.1 N NaOH until a pale pink colour was formed. The acid value was reported as g oleic acid 100 g^−1^ fat [24].

#### 2.2.5. Saponification Value (SV) Determination

About 5.0 g of fat was mixed with 50 mL of 4% KOH solution, and the combination was slowly heated until the sample was fully saponified. After that, it was titrated with 0.5 N HCl while 1% phenolphthalein was used as an indicator. The saponification value was given as mg KOH g^−1^ fat [25].

#### 2.2.6. Total Polar Compounds (TPoC) Determination

A cooking oil tester (Testo 265, Testo SE & Co. KGaA, Lenzkirch, Germany) was used to measure the total polar compounds (TPoC). This instrument provides the content of polar materials with an accuracy of +/−2%. The analysis was carried out by putting the sensor into heated oil and, after about 30 s, reading the temperature and the TPoC content in % from the display. The sensor was calibrated with the manufacturer-supplied calibration oil. Between measurements, the equipment was cleaned with warm water and a neutral detergent. According to Mlcek et al. [26], the allowed limit for TPoC has been established as being 25%.

#### 2.2.7. Total Phenolic Content (TPC) Determination 

TPC content was calculated using Folin–Ciocalteu reagent in accordance with the method of Singleton and Orthofer [27]. Folin–Ciocalteu reagent in a volume of 0.5 mL was combined with 0.1 mL of oil after being diluted 1:10 with distilled water. Before adding 1.7 mL of sodium carbonate solution (20%), the mixture was left to stand at 25 °C for five minutes. After 20 min of incubation with agitation at room temperature, 10 mL of distilled water was added to the mixture and the absorbance was measured at λ = 735 nm. The outcomes were stated as mg of gallic acid kg^−1^ of fat.

#### 2.2.8. GC Analysis

In a round-bottomed flask, 2 mL of sample was added, along with 20 mL of sulphuric acid methanol solution and three pieces of porous porcelain. The flask was placed in a water bath and boiled for approximately 60 min with a reflux cooler attached. Clarifying the solution and observing that there are no longer any fat globules indicated that the reaction was complete. The flask’s contents were allowed to cool to room temperature and then was quantitatively passed in a separatory funnel using 20 mL of water. Twenty millilitres of heptanes were used in two steps of the methyl esters’ extraction. Following the addition of the extracts into a second separatory funnel, the extracts were thoroughly cleaned with 20 mL of water using methyl orange to ensure that all traces of sulphuric acid were absent.

The extracts were filtered into a flask after being dehydrated by the addition of anhydrous sodium sulphate. Solvent traces were eliminated from the sample by nitrogen blowing after the solvent was distilled out in a water bath under vacuum. Following the collection of methyl esters in 1 mL of hexane, 1 μL of the sample was put into the gas chromatograph. A Shimadzu GC-17 A gas chromatograph (Shimadzu, Tokyo, Japan) in conjunction with a flame ionization detector was used to evaluate the composition of fatty acids. Alltech AT-Wax (60 m × 0.32 mm × 0.5 μm) is the gas chromatography column and stationary phase (polyethylene). At 147 kPa of pressure, helium was used as carrier gas, and the injector and detector temperatures were set to 260 °C. The program of the oven was the following: 70 °C for 2 min, after which the temperature was increased to 150 °C with a gradient of 10 °C min^−1^, for a 3 min level, then the temperature was raised to 235 °C with a gradient of 4 °C min^−1^. Comparing the results with standards allowed for the identification and measurement of fatty acids. Results were expressed as g 100 g^−1^ [24].

#### 2.2.9. Microscopic Examination

An Optika-B290 microscope with a tablet was used for the microscopic investigation (Optika, Ponteranica, Italy). Technical characteristics were the following: binocular, 360° rotating and 30° inclined; built-in 3.1 MP camera; interpupillary distance from 48 to 75 mm; vernier scale on the two axes, accuracy: 0.1 mm; double layer rackless mechanical sliding stage, 150 × 139mm, 75 × 33 mm X-Y movement range; objectives: N-PLAN 4×/0.10, N-PLAN 10×/0.25, N-PLAN 40×/0.65, N-PLAN 100×/1.25.

#### 2.2.10. Statistical Analysis

The impact of antioxidant addition and heating temperature on the physicochemical parameters of lard and goose fat were examined using factorial ANOVA with the General Linear Model in Minitab 16.1.0 (LEAD Technologies, Inc., Charlotte, NC, USA). Each sample was examined using three replications. Tukey’s honest significance test was performed at a level of 95% confidence (*p* < 0.05). The degree of connection between chemical parameters was estimated using Pearson’s correlation (α = 0.05) and two-tailed probability values.

## 3. Results and Discussion

The goal of this study was to determine the impact of 0.01% burdock extract added in two alimentary fats (lard and goose), which are commonly used in Romania for culinary cooking, in terms of their oxidative stability and rates of deterioration at various heating temperatures (110, 130, 150, 170, 190, and 210 °C). Chemical analysis results and fatty acid composition for fats and 0.01%-additivated fats upon heating are presented in Table 1, Table 2, Table 3 and Table 4. All the measured parameters were significantly impacted by the heating temperature, with IV and SV being the least affected. PV, TBARS, and TPoC were significantly (*p* < 0.01) influenced by the additive × heat treatment interaction and significantly (*p* < 0.001) affected by the fat type × additive × heat treatment interaction. The first- and second-degree interactions had a significant (*p* < 0.05) effect on IV and a significant (*p* < 0.01) impact on AV (Table 1).

### 3.1. Effect of Additivation and Heat Treatment on Peroxide Value (PV)

The number of peroxides generated during fat oxidation is shown by the peroxide value, which quantifies the primary lipid oxidation. It has been shown that lipid oxidation products such peroxides, free radicals, malonaldehyde, and other products of cholesterol oxidation promote atherosclerosis and coronary heart disease [28]. The peroxide index values were lower in unheated animal fats. The effects of additivation and heating temperature on the peroxide value were statistically significant (*p* < 0.001) (Table 1). The lowest PV was found in lard (Table 2), and the highest was observed in goose fat (Table 3). 

The non-additivated fat had the highest peroxide index level, independent of the heating temperature. The highest intensity of peroxide production was observed at 180 °C, and above this temperature, decomposition was observed. This pattern could be explained by the fact that peroxides are not heat-resistant and that high temperatures reduce their concentration. Peroxides are characteristic during early heating and decompose at high temperatures. The tendency for the peroxide value (PV) in non-additivated animal fats to decrease could be caused by the melting pretreatment’s hydroperoxide-generating effect, which could have been broken down by heating. The fat samples with 0.01% burdock extract presented lower values for PV compared to non-additivated samples. Therefore, a lower number of primary oxidation compounds were determined in burdock additivated fats.

Szabó et al. [29] state that peroxide breakdown takes place at a critical temperature between 190 and 200 °C. This is supported by our data, which show that peroxide generation peaked at 190 °C and that breakdown predominated above this temperature. The PVs of the lard and goose fat samples were significantly affected (*p* < 0.001) by the application of antioxidants (Table 1). According to the results, pork fat produced less peroxides when heated, and therefore, was more stable. The oxidation of both fats occurs at higher heating temperatures, leading to a greater breakdown of polyunsaturated fatty acids and the creation of oxidation products. The development of oxidation products was greatly delayed by the addition of burdock extract in a proportion of 0.01%; however, because of an increased rate of initiation reactions, antioxidant activity decreased with temperature.

A rise in PV during heating or frying of fats and oils has been reported by several researchers [30,31,32]. A study was conducted utilizing the Oxipres apparatus to examine temperature’s effect on the antioxidant activity of α- and δ-tocopherol in pork lard [33]. The research demonstrated that the activity of α-tocopherol remained constant between 80 and 110 °C and decreased with rising temperatures. Pop and Mihalescu [24] investigated the stability of goose, duck, and chicken fats with the addition of natural antioxidants (α-tocopherol and citric acid) during refrigerated and frozen storage. The researchers showed that the results for peroxide value and the thiobarbituric acid-reactive substances test increased with temperature. Furthermore, as storage time increased, the variations in PV and TBARS also increased. The thermal oxidation of tallow was investigated by Song et al. [34], who examined the volatile chemicals produced under various oxidation circumstances, the peroxide value, acid value, and the p-anisidine value (p-AV). The researchers found that, when heated at 140 °C for two hours, the levels of beef-flavour precursors, including hexanal, (E,E)-2,4-decadienal, 1-octen-3-ol, and (E,E)-2,4-heptadienal, achieved their highest value. PV and p-AV were both at high levels while AV was relatively low at the same temperature. The impact of natural and synthetic antioxidants and synergists (green tea extract, rosemary extract, sage extract, α-tocopherol, tocopherol mixture, propyl gallate, citric acid, caffeic acid, ascorbic acid, rosmarinic acid) on the oxidative stability of goose fat was examined using the Schaal Oven test [35]. The results showed that, in goose fat, green tea extract has the strongest antioxidant activity, but rosemary extract in combination with a synergist showed a greater fat-protective factor against oxidation than that of pure rosemary extract. Also, when a blend of tocopherols was used instead of sage extract, the fat was stabilized more effectively.

### 3.2. Effect of Additivation and Heat Treatment on Thiobarbituric Acid-Reactive Substances Test (TBARS)

The quantity of secondary oxidation compounds (ketones, aldehydes, or other matrix compounds) present in the fat sample is measured by TBARS. As for the peroxide value, the values of thiobarbituric acid-reactive compounds increased significantly (*p* < 0.001) in all fats with increasing heating temperatures, but decreased with the addition of burdock extract in a proportion of 0.01%. The results for TBARS increased from 0.84 to 7.62 mg MDA kg^−1^ in pork lard without antioxidant and to 6.38 mg MDA kg^−1^ in burdock-additivated lard exposed to heating at 210 °C; from 1.16 to 8.52 mg MDA kg^−1^ in goose fat without antioxidant and to 7.23 mg MDA kg^−1^ in burdock-additivated goose fat subjected to heating at 210 °C. The TBARS value of non-additivated goose fat heated to 210 °C was higher than the permissible limit of 8 mg MDA kg^−1^ fat. TBARS was significantly affected (*p* < 0.001) by the additivation, type of fat, and heating temperature (Table 1). Regardless of the heating temperature, the highest level of TBARS was detected in fat without an antioxidant. There were significant positive correlations between PV and TBARS in both goose fat (*r* = 0.93; *p* < 0.001) and lard (*r* = 0.96; *p* < 0.001). As for the peroxide value, the values of thiobarbituric acid-reactive compounds increased significantly (*p* < 0.001) in all fats with increasing heating temperatures, but decreased with the addition of burdock extract in a proportion of 0.01%. This is consistent with earlier research on chicken and duck fat, which found that temperature affects the development of TBARS [6].

The potential of some phenolic compounds to prevent the oxidation of butter was evaluated by Soulti and Roussis [36]. Thiobarbituric acid-reactive substances and peroxide values were monitored during butter heating at 50 °C and at 110 °C. The results showed that butylated hydroxyanisole at 200 mg L^−1^ was equally efficient at suppressing butter oxidation at 50 °C as caffeic acid, gallic acid, and catechin at 80 mg L^−1^. Furthermore, gallic acid was more efficient than butylated hydroxyanisole at inhibiting the oxidation of butter at 110 °C. Moslavac et al. [37] studied the effect of natural antioxidants (sage extract, rosemary extract, olive pomace extract, *α*-tocopherol, mixture tocopherol) on the oxidative stability of pork fat using the sustainability test at 98 °C. The results showed that rosemary extract more effectively protected pork fat against oxidation compared to the other antioxidants we tested. It was reported that heating chicken fat at 70 °C and storing it for two and four days significantly increased the amount of lipid oxidation, as shown by TBARS. When heating temperature increased, there was a significant (*p* < 0.05) decreased in both monounsaturated and polyunsaturated fatty acids [5]. 

The oxidative stability of duck, chicken, pork, and bovine fats was evaluated during a period of 90 day of storage at 60 °C [11]. The study showed that the TBARS value was significantly higher in duck fat at the beginning of the storage period (10 to 40 days), and that the value in chicken fat was relatively higher than in pork and bovine fats (10 to 20 days). The results also showed that the higher rate of lipid peroxidation in chicken and duck fats was due to the higher contents of PUFAs which were more susceptible to oxidation. The lower TBARS value determined in bovine fat was due to the higher SFAs content which are less vulnerable to lipid oxidation. Also, the PV levels which were higher in duck, chicken, and swine fat have caused the higher TBARS values. The unstable peroxides can be the result of polyunsaturated fatty acids decomposition; therefore; researchers suggested the addition of antioxidants to duck and chicken fats for improving their oxidative stability during storage.

### 3.3. Effect of Additivation and Heat Treatment on Iodine Value (IV)

The heating temperature significantly (*p* < 0.01) reduced the iodine index values for all types of fats. A decrease in iodine values from 74.7 to 66.1 g I_2_ 100 g^−1^ was observed in pork lard without an antioxidant and to 68.4 g I_2_ 100 g^−1^ in burdock-additivated lard exposed to heating at 210 °C; from 86.8 to 77.4 g I_2_ 100 g^−1^ in goose fat without an antioxidant and to 79.3 g I_2_ 100 g^−1^ in burdock-additivated goose fat exposed to heating at 210 °C. IV was significantly (*p* < 0.01) influenced by the fat type and heating temperatures, and significantly (*p* < 0.05) influenced by the additivation. Fatty acid unsaturation is reduced, as indicated by the decrease in IV levels. The decrease in iodine value during heating indicated a higher rate of oxidation, which may be ascribed to double bond oxidation and polymerization processes. Strong negative correlations between IV and TBARS were determined in lard (*r* = −0.98; *p* < 0.001) and goose fat (*r* = −0.96; *p* < 0.001). In the study conducted by Farhooshi and Moosavi [38], fish crackers were fried at 180 °C for 5 h per day for four consecutive days using an ethanolic citrus peel extract mixed to palm olein at a 0.2% concentration. Iodine value, totox value, peroxide value, and viscosity analyses showed that the orange peel extract had strong antipolymerization and antioxidant properties [38]. Gharby et al. [39] studied the effect of heat treatment on vegetable fats, and a slight decrease in iodine value was observed after 3 h of heating at 180 °C. 

### 3.4. Effect of Additivation and Heat Treatment on Acid Value (AV)

When assessing the hydrolysis extension in fats during the heat process, free acidity is an analytical parameter that is utilized. An increase in this parameter, which is a direct result of hydrolysis and a key marker of the chemical degradation of fat, denotes a larger concentration of free fatty acids in animal fat. At the highest temperatures, the rate of growth in the AV of the fats tended to slow down. The AV increased in parallel with the intensity of heating. The non-additivated fat had the highest AV level, regardless of the heating temperature. Heating at 210 °C produced the largest amount of free fatty acids. Acid value increased from 0.25 to 0.94 g oleic acid 100 g^−1^ in pork lard without an antioxidant and to 0.82 g oleic acid 100 g^−1^ in burdock-additivated lard exposed to 210 °C during heating; from 0.22 to 0.88 g oleic acid 100 g^−1^ in goose fat without an antioxidant and to 0.71 g oleic acid 100 g^−1^ in burdock-additivated goose fat exposed to heating at 210 °C. Strong positive correlations between AV and PV were found in lard (*r* = 0.98; *p* < 0.001) and goose fat (*r* = 0.96; *p* < 0.001). Acid value was significantly (*p* < 0.01) influenced by the fat type and additivation, and significantly (*p* < 0.001) influenced by heating temperature. A rise in the acid value of goose fat and Mangalica-pig lard exposed to heating at seven temperatures (140, 150, 160, 165, 170, 175, and 180 °C) was reported by Szabó et al. [29]. Thermooxidative structural changes in fats were investigated by Gertz et al. [28] at ambient temperatures, under a frying temperature of 170 °C, and under accelerated conditions using 110 °C. The results of the study indicated that the iodine value and the Rancimat test at 110 °C had a strong negative correlation, whereas the anisidine and acid values showed a weak correlation.

### 3.5. Effect of Additivation and Heat Treatment on Saponification Value (SV)

The lipids’ average molecular weights are indicated by the saponification value. Oxidative rancidity increases SV by oxidizing fatty acids, which forms aldehydes and ketones. Strong positive correlations between AV and SV were determined in pork lard (*r* = 0.94; *p* < 0.001), and goose fat (*r* = 0.97; *p* < 0.001). All treatments had a significant (*p* < 0.01) impact on saponification value. For both lard and goose fat, saponification value increased significantly (*p* < 0.01) with heating temperature. In a model of frying at 180 °C, Patel et al. [40] found that the stability of clarified butterfat supplemented with 0.5% commercial steam-distilled coriander extract and oleoresin has been significantly improved. The steam-distilled extract performed better than the oleoresin, according to the measurements for conjugated dienes, peroxide value, thiobarbituric acid value, saponification value, and the Rancimat at 120 °C.

### 3.6. Effect of Additivation and Heat Treatment on Total Polar Compounds (TPoC)

During heating, peroxides and hydroperoxides produce total polar compounds, which include alcohol, nonvolatile products, ketones, aldehydes, and short-chain fatty acids. In every sample that was subjected to heating, the TPoC values were below the limit of 25%. In comparison to pork lard, goose fat exhibited greater values of TPoC, which increased in parallel with the heating temperature. The heating temperature had a significant (*p* < 0.001) effect on the TPoC values, as shown in Table 1. The formation of TPoC was significantly (*p* < 0.01) reduced by the addition of burdock extract at a concentration of 0.01%. Qiuyu et al. [41] studied the thermal behaviour and antioxidant effect of rosemary extract on chicken fat. A synchronization was seen between the reduction in antioxidant capacity and weight loss, as the researchers discovered that rosemary extract was stable below 200 °C and that phenolics were resistant below 130 °C.

### 3.7. Effect of Additivation and Heat Treatment on Total Phenolic Content (TPC)

The heat treatment at 210 °C resulted in a decrease in total phenolic content from 148 to 96 mg gallic acid kg^−1^ in pork lard and from 163 to 102 mg gallic acid kg^−1^ in additivated lard under heating at 210 °C (Table 2). Total phenolic content decreased from 164 to 115 mg gallic acid kg^−1^ in goose fat exposed to heating at 210 °C and from 185 to 139 mg gallic acid kg^−1^ in additivated goose fat under heating at 210 °C (Table 3). Total phenolic content was reduced by 35% in pork lard without an antioxidant and by 23% in burdock-additivated lard under heating. For goose fat, the reduction was about 30% in non-additivated fat, and about 19% in burdock-additivated fat under heating. According to Xueqi et al. [42], most of the squalene in olive oil remained intact even after heating at 220 °C. A considerable quantity of total phenols and specific phenols, including oleocanthal, remained after heating the oil at 121 °C for 10 and 20 min. Oleocanthal had the highest temperature tolerance among the studied phenols, hydroxytyrosol the lowest, while tyrosol exhibited a smaller change with various heating techniques. The study reported a loss of total phenol content of 40.80% after 10 min of heating at 121 °C, while 78.41% of total phenol loss was observed after 10 min of heating at 220 °C.

The total phenolic content in the 70%-ethanolic extract of *A. lappa* leaves was 97.49 mg g^−1^ [43], whereas Lee et al. [44] determined that the total phenolic content of cultivated burdock was higher in the roots than in the leaves (137 and 41.4 mg 100 g^−1^ dry material, respectively). Burdock leaf fractions’ effectiveness in preserving meat was assessed [45]. The findings demonstrated that *Salmonella Typhimurium* and *Escherichia coli* growth and biofilm formation were greatly suppressed by burdock leaf fraction. When compared to the pork without treatment, the shelf life of pork treated with burdock leaf fractions was increased by six days, and the pork’s sensory qualities were clearly improved. Chemical composition analysis indicated that the burdock-leaf fraction consisted of caffeic acid, chlorogenic acid, *p*-coumaric acid, cynarin, rutin, luteolin, crocin, arctiin, and quercetin. The burdock-leaf fraction was shown to be a promising natural preservative for pork.

### 3.8. Effect of Heat Treatment on Fatty Acid Composition

The fatty acid contents of dietary fats (non-additivated and additivated with burdock extract) subjected to heating at 210 °C are presented in Table 4. Among the fatty acids, palmitic (C16:0), palmitoleic (C18:1), and oleic (C16:1) are the most popular in pork lard; palmitoleic (C18:1) and oleic (C16:1) are the most popular in goose fat. Saturated fatty acids (SFAs) are predominant in lard, whereas monounsaturated fatty acids (MUFAs) are predominant in goose fat. Goose fat had the highest concentration of polyunsaturated fatty acids (PUFAs), followed by the lard. For lard without an antioxidant, total SFAs increased from 4.20 (g 100 g^−1^ fat) in the control to 4.96 in lard heated to 210 °C, and for non-additivated goose fat from 2.32 (g 100 g^−1^ fat) in the control to 2.58 in goose fat heated to 210 °C. These increases were attributed to the high-temperature heating treatment. The total PUFAs decreased from 3.76 (g 100 g^−1^ fat) in the control to 2.61 in goose fat without antioxidant exposed to heating at 210 °C, and from 2.29 in the control to 1.42 in lard without antioxidant exposed to heating at 210 °C. Heat treatment at 210 °C did not significantly modify the composition of stearic and palmitic acids, while a considerable rise in the content of heptadecanoic and myristic acids was detected. Unsaturated fatty acids are altered by heat treatment of fats, they may isomerize from their cis to transform. During thermal treatment, such formation of trans fatty acids has been observed for chicken fat [46]. The heating temperature had a significant effect on total MUFAs in both lard and goose fat (to a greater extent in fat without an antioxidant). Considering that highest level of PV was found in goose fat, the oxidation of monounsaturated fatty acids may be the cause of the decrease in total MUFA. Palmitoleic acid (C18:1) significantly decreased in goose fat exposed to heating at 210 °C (with 0.39 units) and in 0.01% burdock-additivated fat exposed to heating at 210°C (with 0.21 units). From the category of polyunsaturated fatty acid, linolenic acid (C18:3), decreased significantly in goose fat exposed to heating at 210 °C (with 0.17 units) and burdock-additivated fat exposed to heating at 210 °C (with 0.12 units). The strongest correlation between heating temperature and MUFA was found in goose fat (*r* = 0.94), followed by the pork lard (*r* = 0.86). The oxidation of PUFA and MUFA was reduced by the addition of burdock extract in a 0.01% proportion.

Our results are in agreement with those reported by Niu et al. [46] who noticed an increase in total saturated fatty acid and a decrease in mono- and polyunsaturated fatty acids during thermal treatment of lard. It was reported that lard oil contained higher percentages of palmitic (35.25%) and stearic (16.99%) acids, and during thermal treatment, there were changes in the distribution of fatty acids with an increase in saturated fatty acid and a decrease in polyunsaturated fatty acids which was the most affected fraction [12]. The content of total fatty acids in chicken fat heated at different temperatures was reported [47]. Oleic (C18:1), linoleic (C18:2), and palmitic (C16:0) acids were found to be the most abundant fatty acids, and almost 60% of the total fatty acids were unsaturated fatty acids. As the heating temperature rose, the amount of saturated fatty acids generally increased while the amounts of monounsaturated and polyunsaturated fatty acids significantly (*p* < 0.05) decreased [47]. Changes in the fatty acid profile of the subcutaneous and intramuscular fat as well as a sensory evaluation of goose packaged in a modified atmosphere under different conditions and under refrigeration storage at 4 °C were investigated [8]. The results showed that a highly oxygen-modified atmosphere had a negative impact on the quality of goose meat due to a reduction in polyunsaturated fatty acids and an increase in saturated fatty acids, which indicates a significant reduction in its nutritional value. It was observed that the fatty acid composition of goose meat remains unchanged for 11 days of storage, as long as a vacuum is used to reduce the amount of oxygen exposure. Vacuum packaging proved to be a more effective technique for maintaining the fatty acid profile and flavour of the goose meat.

The fatty acid composition of duck, chicken, pork, and bovine fats was analysed by Shin et al. [11]. The major fatty acids identified in duck fat were oleic acid (C18:1), palmitic acid (C16:0), stearic acid (18:0), linoleic acid (C18:2), and arachidonic acid (C20:4). For chicken fat, oleic acid (C18:1) was the major unsaturated fatty acid, followed by linoleic acid (C18:2), palmitic acid (C16:0), and stearic acid (C18:0), but only bovine fat showed more stearic acid (C18:0) than linoleic acid (C18:2). The study showed that total SFAs and PUFAs were found to be significantly (*p* < 0.05) different across the groups. Duck fat showed the highest content of PUFAs and the lowest content of SFAs, while bovine fat presented the lowest PUFA content and the highest SFA content. The PUFA/SFA ratio of duck fat was approximately two times higher than that of bovine fat.

### 3.9. Effect of Heat Treatment on Microscopic View

The aim of the microscopic examination was to monitor changes that occur in different stages of the oxidation process in animal fats at the microscopic level, relating them to their physical structure. 

The microscope view of the unheated animal fats showed fat cells arranged in chains, which were compact and well highlighted. Along with the increase in the exposure time, fat cells were destroyed, weakly highlighted, and irregularly distributed. At higher temperatures, fat cells were almost completely destroyed and barely visible (Figure 1 and Figure 2). In our analysis of the literature, we have found no study that investigates the oxidation process through a microscopic examination, therefore microscopic analysis may represent a new method for the evaluation of oxidative processes and should be further studied. Animal lipids were well protected from oxidation by the burdock extract, which demonstrated its efficacy as an antioxidant; it may be used to monitor the fats oxidation and to estimate their shelf-life stability.

## 4. Conclusions

Heating temperature, type of fat, and antioxidant addition affected the rate of lipid oxidation in dietary animal fats. The results showed that pork lard was more resistant to heating, generating smaller amounts of peroxides. Peroxide value, thiobarbituric acid-reactive substances, acid value, total polar compounds and total phenolic content were significantly influenced by the heating temperature. The contents of primary and secondary oxidation compounds were lower in the samples with an added antioxidant. Heat treatment induced several chemical reactions such as oxidative, hydrolytic, and polymerization reactions, which modify the fatty acid composition of the heated fats. The heating temperature had a significant effect on total MUFAs in both lard and goose fat (to a greater extent in fat without antioxidant). Microscopic examination indicated differences between fresh and heated fats. It may represent a new method for the evaluation of oxidative processes, thus should be further studied. The addition of burdock extract in concentration of 0.01% significantly inhibited lipid oxidation in dietary lard and goose fat exposed to heating at varying temperatures, therefore it can be used as a potential antioxidant to increase fats’ stability.

## Figures and Tables

**Figure 1 foods-13-00304-f001:**
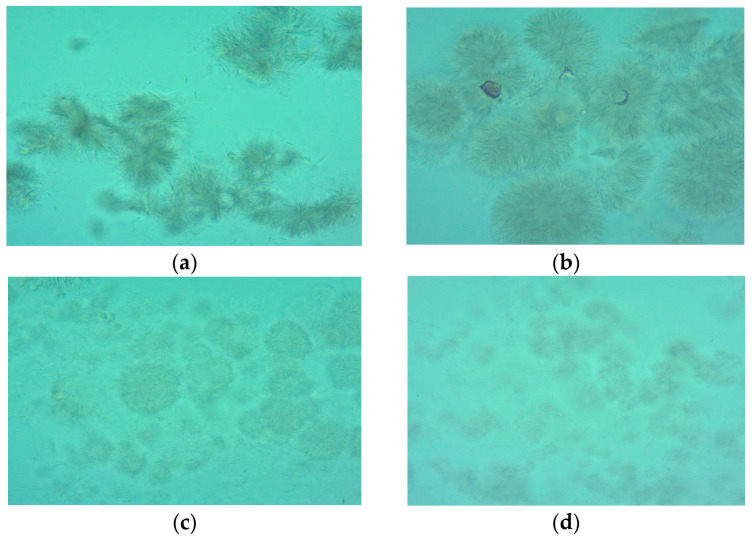
Microscope view of pork lard subjected to heating. (**a**)—microscope view of unheated lard; (**b**)—microscope view of lard subjected to heating at 130 °C; (**c**)—microscope view of lard subjected to heating at 150 °C; (**d**)—microscope view of lard subjected to heating at 170 °C; (**e**)—microscope view of lard subjected to heating at 190 °C; (**f**)—microscope view of lard subjected to heating at 210 °C.

**Figure 2 foods-13-00304-f002:**
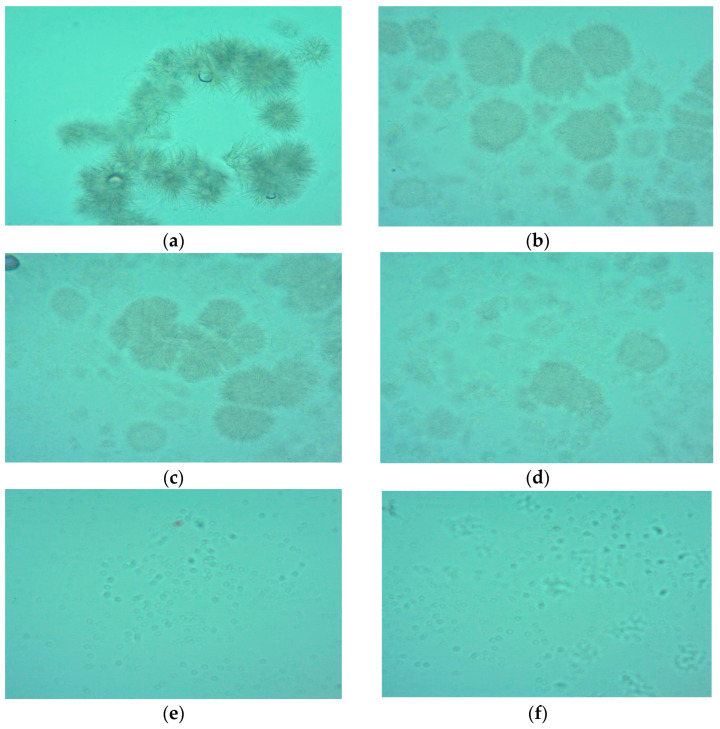
Microscope view of goose fat subjected to heating. (**a**)—microscope view of unheated goose fat; (**b**)—microscope view of goose fat subjected to heating at 130 °C; (**c**)—microscope view of goose fat subjected to heating at 150 °C; (**d**)—microscope view of goose fat subjected to heating at 170 °C; (**e**)—microscope view of goose fat subjected to heating at 190 °C; (**f**)—microscope view of goose fat subjected to heating at 210 °C.

**Table 1 foods-13-00304-t001:** The impact of the fat type, additivation, heat treatment, and their first- and second-degree interactions, on PV (meq O_2_ kg^−1^ fat), TBARS (mg MDA kg^−1^ fat), IV (g I_2_ 100 g^−1^ fat), AV (g oleic acid 100 g^−1^ fat), SV (mg KOH g^−1^ fat), TPoC (%), and TPC (mg gallic acid kg^−1^ fat).

Factor	PV	TBARS	IV	AV	SV	TPoC	TPC
Fat type							
Lard	4.02 ^a^	3.15 ^a^	81.2 ^b^	0.29 ^a^	195 ^b^	1.4 ^a^	164 ^a^
Goose fat	5.19 ^b^	5.38 ^b^	82.5 ^a^	0.20 ^b^	198 ^a^	1.6 ^b^	168 ^b^
*p*	<0.01 **	<0.001 ***	<0.01 **	<0.01 **	<0.01 **	<0.05 *	<0.05 *
Additivation							
Non-additivated	5.60 ^a^	4.56 ^a^	82.7 ^b^	0.45 ^a^	198 ^a^	2.7 ^a^	165 ^a^
Additivated with 0.01% burdock extract	4.40 ^b^	3.20 ^b^	83.2 ^a^	0.39 ^b^	195 ^b^	1.5 ^b^	184 ^b^
*p*	<0.001 ***	<0.001 ***	<0.05 *	<0.01 **	<0.01 **	<0.01 **	<0.01 **
Heat treatment							
Unheated	1.95 ^f^	0.96 ^g^	84.7 ^a^	0.21 ^f^	195 ^e^	1.4 ^a^	164 ^a^
110 °C	2.46 ^e^	1.14 ^f^	84.5 ^a^	0.28 ^e^	196 ^de^	1.9	161 ^b^
130 °C	3.16 ^d^	1.87 ^e^	84.1 ^b^	0.35 ^d^	197 ^d^	2.7 ^ab^	157 ^c^
150 °C	5.47 ^c^	2.98 ^d^	83.2 ^b^	0.41 ^c^	198 ^c^	3.8 ^a^	149 ^d^
170 °C	7.64 ^b^	4.56 ^c^	81.5 ^c^	0.63 ^b^	200 ^bc^	5.3 ^ab^	142 ^e^
190 °C	8.84 ^a^	6.48 ^b^	80.4 ^c^	0.78 ^ab^	201 ^b^	8.2 ^ab^	135 ^f^
210 °C	8.70 ^a^	7.64 ^a^	78.6 ^d^	0.86 ^a^	204 ^a^	12.4 ^b^	118 ^g^
*p*	<0.001 ***	<0.001 ***	<0.01 **	<0.001 ***	<0.01 **	<0.001 ***	<0.001 ***
Additive * Heat treatment							
Additive * unheated	1.83 ^e^	0.84 ^d^	76.8 ^a^	0.25 ^d^	194 ^b^	1.1 ^a^	172 ^a^
Additive * 110 °C	2.54 ^d^	1.16 ^cd^	76.3 ^a^	0.27 ^cd^	194 ^b^	1.8 ^ab^	170 ^a^
Additive * 130 °C	2.96 ^cd^	1.85 ^c^	75.5 ^b^	0.32 ^c^	195 ^ab^	2.7 ^b^	162 ^b^
Additive * 150 °C	3.48 ^c^	2.17 ^bc^	75.1 ^b^	0.39 ^c^	195 ^ab^	3.5 ^bc^	154 ^bc^
Additive * 170 °C	6.53 ^bc^	3.61 ^b^	74.3 ^c^	0.46 ^bc^	195 ^ab^	4.9 ^c^	141 ^c^
Additive * 190 °C	7.12 ^b^	5.28 ^ab^	74.1 ^c^	0.57 ^b^	196 ^a^	5.8 ^cd^	132 ^cd^
Additive * 210 °C	8.51 ^a^	6.44 ^a^	73.4 ^cd^	0.69 ^a^	196 ^a^	7.2 ^d^	125 ^d^
*p*	<0.01**	<0.01 **	<0.05 *	<0.01 **	≥0.05	<0.01 **	<0.05 *
Fat type * additive * heat treatment							
Lard * additive * unheated	1.72 ^f^	0.82 ^f^	74.3 ^c^	0.25 ^de^	193 ^c^	0.8 ^a^	162 ^bc^
Lard * additive * 110 °C	1.94 ^f^	1.05 ^f^	74.2 ^c^	0.27 ^d^	193 ^c^	1.5 ^b^	157 ^c^
Lard * additive * 130 °C	2.26 ^e^	1.79 ^e^	73.5 ^cd^	0.34 ^c^	194 ^bc^	2.2 ^bc^	144 ^cd^
Lard * additive * 150 °C	3.15 ^d^	2.36 ^de^	72.1 ^cd^	0.46 ^bc^	195 ^b^	3.4 ^c^	132 ^d^
Lard * additive * 170 °C	5.03 ^bc^	3.09 ^d^	71.9 ^cd^	0.65 ^b^	197 ^ab^	5.1 ^d^	123 ^de^
Lard * additive * 190 °C	5.98 ^bc^	4.26 ^c^	69.7 ^d^	0.77 ^ab^	198 ^ab^	6.7 ^e^	114 ^e^
Lard * additive * 210 °C	6.85 ^b^	6.11 ^b^	68.7 ^d^	0.81 ^a^	199 ^a^	8.2 ^ef^	103 ^ef^
Goose * additive * unheated	2.13 ^e^	1.12 ^ef^	86.3 ^a^	0.21 ^de^	194 ^bc^	1.4 ^b^	187 ^a^
Goose * additive * 110 °C	2.38 ^e^	1.32 ^ef^	86.0 ^a^	0.24 ^d^	195 ^b^	1.8 ^b^	182 ^ab^
Goose * additive * 130 °C	3.09 ^d^	1.92 ^e^	85.4 ^ab^	0.32 ^c^	195 ^b^	3.5 ^c^	175 ^b^
Goose * additive * 150 °C	4.31 ^c^	3.85 ^cd^	84.3 ^ab^	0.41 ^bc^	196 ^b^	4.6 ^cd^	166 ^bc^
Goose * additive * 170 °C	6.36 ^b^	4.21 ^c^	82.9 ^b^	0.55 ^bc^	198 ^ab^	7.2 ^e^	152 ^c^
Goose * additive * 190 °C	7.34 ^ab^	6.34 ^b^	81.7 ^b^	0.63 ^b^	199 ^a^	8.6 ^ef^	146 ^cd^
Goose * additive * 210 °C	8.45 ^a^	7.48 ^a^	79.6 ^bc^	0.72 ^ab^	200 ^a^	12.7 ^f^	138 ^d^
*p*	<0.001 ***	<0.001 ***	<0.05 *	<0.01 **	<0.05 *	<0.001 ***	<0.01 **

PV, peroxide value; TBARS, thiobarbituric acid-reactive substances; IV, iodine value; AV, acid value; SV, saponification value; TPoC total polar compounds; TPC, total phenolic content. Values are indicated as mean. Variations in the same column’s letters denote statistically significant differences at *p* < 0.05 (Tukey’s test). Significant differences are denoted by asterisks: * *p* < 0.05; ** *p* < 0.01; *** *p* < 0.001; *p* ≥ 0.05, non-significant.

**Table 2 foods-13-00304-t002:** Monitoring of quality parameters in lard (non-additivated and additivated with 0.01% burdock extract) exposed to heating at different temperatures.

Lard	Heating Temperature	PV (meq O_2_ kg^−1^ Fat)	TBARS (mg MDA kg^−1^ Fat)	IV (g I_2_ 100 g^−1^ Fat)	AV (g Oleic Acid 100 g^−1^ Fat)	SV (mg KOH g^−1^ Fat)	TPoC (%)	TPC (mg Gallic Acid kg^−1^ Fat)
Non-additivated	Unheated	1.76 ^i^ ± 0.02	0.84 ^k^ ± 0.01	74.7 ^a^ ± 0.3	0.25 ^h^ ± 0.02	194 ^e^ ± 0.1	1.1 ^i^ ± 0.3	148 ^g^ ± 0.4
	110 °C	2.24 ^g^ ± 0.03	1.26 ^f^ ± 0.05	74.2 ^a^ ± 0.1	0.31 ^h^ ± 0.07	196 ^d^ ± 0.6	2.3 ^g^ ± 0.5	141 ^f^ ± 0.5
	130 °C	3.18 ^f^ ± 0.01	2.13 ^e^ ± 0.04	73.5 ^b^ ± 0.4	0.42 ^f^ ± 0.02	197 ^d^ ± 0.7	3.7 ^f^ ± 0.4	134 ^e^ ± 0.3
	150 °C	4.85 ^e^ ± 0.04	3.25 ^d^ ± 0.02	71.4 ^c^ ± 0.3	0.59 ^e^ ± 0.05	198 ^c^ ± 0.5	4.8 ^e^ ± 0.6	122 ^d^ ± 0.2
	170 °C	7.02 ^b^ ± 0.07	4.78 ^c^ ± 0.07	70.2 ^d^ ± 0.2	0.78 ^c^ ± 0.04	199 ^c^ ± 0.4	7.9 ^c^ ± 0.5	113 ^c^ ± 0.1
	190 °C	8.79 ^a^ ± 0.05	5.48 ^b^ ± 0.03	68.5 ^e^ ± 0.1	0.84 ^b^ ± 0.03	201 ^b^ ± 0.2	9.5 ^b^ ± 0.4	104 ^b^ ± 0.2
	210 °C	7.65 ^b^ ± 0.06	7.62 ^a^ ± 0.08	66.1 ^f^ ± 0.5	0.94 ^a^ ± 0.02	203 ^a^ ± 0.1	10.7 ^a^± 0.2	96 ^a^ ± 0.4
Additivated with 0.01% burdock extract	Unheated	1.75 ^i^ ± 0.02	0.83 ^k^ ± 0.04	74.5 ^a^ ± 0.4	0.24 ^h^ ± 0.01	194 ^e^ ± 0.3	0.9 ^i^ ± 0.3	163 ^i^ ± 0.2
	110 °C	2.11 ^h^ ± 0.01	1.02 ^g^ ± 0.07	74.1 ^a^ ± 0.5	0.28 ^h^ ± 0.05	194 ^e^ ± 0.4	1.6 ^h^ ± 0.2	156 ^h^ ± 0.3
	130 °C	2.47 ^g^ ± 0.03	1.83 ^f^ ± 0.01	73.4 ^b^ ± 0.3	0.36 ^g^ ± 0.06	195 ^d^ ± 0.8	2.4 ^g^ ± 0.5	145 ^f^ ± 0.5
	150 °C	3.72 ^f^ ± 0.04	2.41 ^e^ ± 0.02	72.4 ^c^ ± 0.2	0.48 ^e^ ± 0.02	196 ^d^ ± 0.5	3.5 ^f^ ± 0.6	131 ^e^ ± 0.4
	170 °C	5.84 ^d^ ± 0.07	3.26 ^d^ ± 0.03	71.7 ^d^ ± 0.1	0.64 ^d^ ± 0.04	198 ^c^ ± 0.6	5.3 ^e^ ± 0.4	124 ^d^ ± 0.1
	190 °C	6.23 ^c^ ± 0.05	4.52 ^c^ ± 0.06	69.5 ^e^ ± 0.6	0.78 ^c^ ± 0.01	200 ^c^ ± 0.2	6.9 ^d^ ± 0.1	112 ^c^ ± 0.2
	210 °C	7.36 ^b^ ± 0.02	6.38 ^b^ ± 0.08	68.4 ^e^ ± 0.1	0.82 ^b^ ± 0.02	201 ^b^ ± 0.4	8.3 ^b^ ± 0.8	102 ^b^ ± 0.4

PV, peroxide value; TBARS, thiobarbituric acid-reactive substances; IV, iodine value; AV, acid value; SV, saponification value; TPoC total polar compounds; TPC, total phenolic content. Values are expressed as mean ± standard deviation of three replicates for each parameter. Variations in the same column’s letters denote statistically significant differences at *p* < 0.05 (Tukey’s test).

**Table 3 foods-13-00304-t003:** Monitoring of quality parameters in goose fat (non-additivated and additivated with 0.01% burdock extract) exposed to heating at different temperatures.

Goose Fat	Heating Temperature	PV (meq O_2_ kg^−1^ Fat)	TBARS (mg MDA kg^−1^ Fat)	IV (g I_2_ 100 g^−1^ Fat)	AV (g Oleic Acid 100 g^−1^ Fat)	SV (mg KOH g^−1^ fat)	TPoC (%)	TPC (mg Gallic Acid kg^−1^ Fat)
Non-additivated	Unheated	2.28 ^i^ ± 0.07	1.16 ^k^ ± 0.03	86.8 ^a^ ± 0.3	0.22 ^h^ ± 0.03	195 ^e^ ± 0.4	1.4 ^i^ ± 0.2	164 ^f^ ± 0.2
	110 °C	2.76 ^g^ ± 0.05	1.54 ^f^ ± 0.02	86.1 ^a^ ± 0.1	0.29 ^g^ ± 0.04	196 ^d^ ± 0.8	2.8 ^g^ ± 0.4	160 ^e^ ± 0.1
	130 °C	3.96 ^f^ ± 0.03	2.85 ^e^ ± 0.03	84.3 ^b^ ± 0.4	0.37 ^f^ ± 0.07	196 ^d^ ± 0.6	4.3 ^e^ ± 0.1	152 ^d^ ± 0.4
	150 °C	5.94 ^e^ ± 0.02	3.91 ^d^ ± 0.06	82.7 ^c^ ± 0.3	0.51 ^d^ ± 0.02	198 ^c^ ± 0.2	5.8 ^e^ ± 0.6	143 ^c^ ± 0.5
	170 °C	8.58 ^d^ ± 0.06	5.74 ^c^ ± 0.04	81.3 ^d^ ± 0.2	0.69 ^c^ ± 0.05	200 ^c^ ± 0.3	8.6 ^c^ ± 0.7	135 ^c^ ± 0.3
	190 °C	9.71 ^a^ ± 0.02	7.25 ^b^ ± 0.01	78.6 ^e^ ± 0.1	0.78 ^b^ ± 0.01	202 ^b^ ± 0.1	10.3 ^b^ ± 0.3	128 ^b^ ± 0.6
	210 °C	8.32 ^c^ ± 0.04	8.54 ^a^ ± 0.07	77.4 ^f^ ± 0.5	0.88 ^a^ ± 0.06	204 ^a^ ± 0.5	15.1 ^a^± 0.9	115 ^a^ ± 0.5
Additivated with 0.01% burdock extract	Unheated	2.15 ^j^ ± 0.03	1.15 ^k^ ± 0.02	86.2 ^a^ ± 0.2	0.22 ^h^ ± 0.04	194 ^f^ ± 0.7	1.5 ^i^ ± 0.1	185 ^i^ ± 0.1
	110 °C	2.31 ^i^ ± 0.04	1.32 ^g^ ± 0.05	86.1 ^a^ ± 0.1	0.25 ^g^ ± 0.03	195 ^e^ ± 0.2	1.9 ^h^ ± 0.4	181 ^h^ ± 0.2
	130 °C	3.14 ^h^ ± 0.02	1.96 ^f^ ± 0.04	85.2 ^b^ ± 0.4	0.31 ^f^ ± 0.02	196 ^d^ ± 0.4	3.6 ^f^ ± 0.6	174 ^g^ ± 0.3
	150 °C	4.35 ^f^ ± 0.01	2.85 ^e^ ± 0.03	84.3 ^c^ ± 0.6	0.43 ^e^ ± 0.05	197 ^e^ ± 0.7	4.8 ^e^ ± 0.5	165 ^f^ ± 0.8
	170 °C	7.26 ^d^ ± 0.06	4.51 ^d^ ± 0.06	82.8 ^d^ ± 0.3	0.56 ^d^ ± 0.07	199 ^c^ ± 0.1	7.1 ^d^ ± 0.3	153 ^d^ ± 0.2
	190 °C	7.83 ^bc^ ± 0.05	6.27 ^c^ ± 0.02	81.4 ^e^ ± 0.5	0.62 ^c^ ± 0.04	200 ^c^ ± 0.4	8.7 ^c^ ± 0.2	145 ^c^ ± 0.4
	210 °C	9.05 ^ab^ ± 0.02	7.23 ^b^ ± 0.01	79.3 ^f^ ± 0.2	0.71 ^b^ ± 0.02	202 ^b^ ± 0.2	12.8 ^b^ ± 0.7	139 ^b^ ± 0.5

PV, peroxide value; TBARS, thiobarbituric acid-reactive substances; IV, iodine value; AV, acid value; SV, saponification value; TPoC total polar compounds; TPC, total phenolic content. Values are expressed as mean ± standard deviation of three replicates for each parameter. Variations in the same column’s letters denote statistically significant differences at *p* < 0.05 (Tukey’s test).

**Table 4 foods-13-00304-t004:** Variations in fatty acid content (g 100 g^−1^ fat) of dietary fats exposed to heating at 210°C.

Fatty Acids	Lard			Goose Fat		
	Non-Additivated (Control)	Non-Additivated Fat Subjected to Heating at 210 °C	Burdock Extract Additivated Fat Subjected to Heating at 210 °C	Non-Additivated (Control)	Non-Additivated Fat Subjected to Heating at 210 °C	Burdock Extract Additivated Fat Subjected to Heating at 210 °C
Myristic (14:0)	0.52 ^a^ ± 0.04	0.73 ^a b^ ± 0.06	0.87 ^b^ ± 0.09	0.31 ^a^ ± 0.10	0.52 ^b^ ± 0.05	0.36 ^a^ ± 0.05
Pentadecanoic (C15:0)	0.65 ^a^ ± 0.06	0.86 ^b^ ± 0.01	0.86 ^b^ ± 0.02	0.52 ^a^ ± 0.14	0.67 ^ab^ ± 0.07	0.58 ^a^ ± 0.04
Palmitic (C16:0)	1.80 ^a^ ± 0.10	1.88 ^a^ ± 0.08	1.84 ^a^ ± 0.13	0.61 ^a^ ± 0.12	0.72 ^a^ ± 0.12	0.62 ^a^ ± 0.12
Palmitoleic (C16:1)	1.18 ^a^ ± 0.13	1.15 ^a^ ± 0.13	1.28 ^a^ ± 0.11	1.85 ^a^ ± 0.03	1.45 ^b^ ± 0.02	1.59 ^b^ ± 0.07
Hexadecadienoic (C16:2)	0.77 ^a^ ± 0.05	0.45 ^b^ ± 0.11	0.36 ^b^ ± 0.04	0.78 ^a^ ± 0.08	0.62 ^a^ ± 0.09	0.61 ^a^ ± 0.06
Hexadecatrienoic (C16:3)	0.56 ^a^ ± 0.02	0.34 ^ab^ ± 0.12	0.21 ^b^ ± 0.09	0.56 ^a^ ± 0.04	0.32 ^b^ ± 0.10	0.48 ^a^ ± 0.08
Hexadecatetranoic (C16:4)	0.19 ^a^ ± 0.06	0.17 ^a^ ± 0.01	0.13 ^a^ ± 0.01	0.59 ^a^ ± 0.11	0.37 ^ab^ ± 0.15	0.54 ^a^ ± 0.13
Heptadecanoic (C17:0)	0.86 ^a^ ± 0.03	1.03 ^ab^ ± 0.05	1.17 ^b^ ± 0.05	0.51 ^a^ ± 0.05	0.64 ^ab^ ± 0.06	0.57 ^a^ ± 0.09
Stearic (C18:0)	0.93 ^a^ ± 0.09	1.07 ^a^ ± 0.02	1.09 ^a^ ± 0.15	0.32 ^a^ ± 0.09	0.33 ^a^ ± 0.08	0.31 ^a^ ± 0.11
Oleic (C18:1)	2.24 ^a^ ± 0.14	1.86 ^ab^ ± 0.09	2.16 ^a^ ± 0.13	1.49 ^a^ ± 0.13	1.22 ^ab^ ± 0.03	1.46 ^a^ ± 0.05
Linoleic (C18:2)	0.41 ^a^ ± 0.11	0.33 ^ab^ ± 0.07	0.36 ^a^ ± 0.10	0.66 ^a^ ± 0.04	0.45 ^b^ ± 0.08	0.50 ^b^ ± 0.08
Linolenic (C18:3)	0.27 ^a^ ± 0.12	0.18 ^ab^ ± 0.03	0.22 ^a^ ± 0.11	0.56 ^a^ ± 0.06	0.42 ^b^ ± 0.01	0.47 ^ab^ ± 0.01
Stearidonic acid (C18:4)	0.08 ^a^ ± 0.05	0.07 ^a^ ± 0.06	0.07 ^a^ ± 0.07	0.79 ^a^ ± 0.15	0.54 ^b^ ± 0.04	0.84 ^a^ ± 0.13
Total FA	10.58 ^a^ ± 0.88	10.07 ^b^ ± 0.67	10.75 ^a^ ± 1.09	9.58 ^a^ ± 0.95	8.29 ^b^ ± 0.90	8.41 ^b^ ± 1.03
Total SFA	4.20 ^a^ ± 0.32	4.96 ^ab^ ± 0.06	4.93 ^ab^ ± 0.44	2.32 ^a^ ± 0.32	2.58 ^ab^ ± 0.32	2.38 ^a^ ± 0.40
Total MUFA	4.07 ^a^ ± 0.27	3.65 ^b^ ± 0.22	4.03 ^a^ ± 0.23	2.37 ^a^ ± 0.16	3.08 ^c^ ± 0.05	2.63 ^b^ ± 0.12
Total PUFA	2.29 ^a^ ± 0.30	1.42 ^c^ ± 0.40	1.79 ^b^ ± 0.42	3.76 ^a^ ± 0.48	2.61 ^c^ ± 0.47	3.40 ^b^ ± 0.50

FA, fatty acids; SFA, saturated fatty acids; MUFA, monounsaturated fatty acids; PUFA, polyunsaturated fatty acids. Values are expressed as mean ± standard deviation of three replicates for each parameter. For each type of animal fat, different letters in the same row indicate statistically significant differences at *p* < 0.05 (Tukey’s test).

## Data Availability

Data is contained within the article.

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
