# Peer review of "The Antioxidant Effect of Burdock Extract on the Oxidative Stability of Lard and Goose Fat during Heat Treatment"

_foods, 2024, doi:10.3390/foods13020304_

Round 1

Reviewer 1 Report

Comments and Suggestions for Authors

Regarding the manuscript entitled The Antioxidant Effect of Burdock Extract on the Oxidative Stability of Lard and Goose Fat During Heat Treatment

L20-21. Delete, it should be in the conclusion section.

L23. Rephrase.

L24. Add a conclusion sentence with a recommendation to the readers.

L89. Add hypothesis.

L90-93. This paragraph should be above the aims.

L101. Deleted small amounts.

L107. Ref to the method of fat extraction.

L120. Add company name, city, and country to all devices and chemicals involved in the manuscript. Please double-check others throughout the manuscript.

Results and discussion, add numbering to sub-headings.

Table 1. What about the interaction between additive and heat treatment or fat type*heat treatment*additive? I think three factors should be considered in statistical analysis and their interactions.

Table 2 and Table 3. Why did the authors separate between two fat types (lard and goose)? Otherwise, they should choose one fat type and examine it. You have three factors in your design fat type, temperature, and additive.

L84. Only after heat treatment is included in Table 4 or double-check the titles within the table.

Discussion section the mechanism of action is lacking, or interpretation of the findings is weak. Please revise your discussion section focusing on the mechanism of action.

The conclusion section should contain a strong message or recommendation to the readers without repeating your findings and delete p values in the conclusion.

Comments on the Quality of English Language

moderate editing

Author Response

Regarding the manuscript entitled The Antioxidant Effect of Burdock Extract on the Oxidative Stability of Lard and Goose Fat During Heat Treatment

  1. L20-21. Delete, it should be in the conclusion section.

Answer: We have deleted L20-21 and included them in conclusion section.

  1. L23. Rephrase.

Answer: Line 23 was rephrased (new line 21-22).

  1. L24. Add a conclusion sentence with a recommendation to the readers.

Answer: The clonclusion sentence was added at the end of the abstract (lines 22-24).

  1. L89. Add hypothesis.

Answer: The hypothesis was added at the end of introduction section.

  1. L90-93. This paragraph should be above the aims.

Answer: The mentioned paragraph was inserted above the aim according to your suggestion (lines 96-99).

  1. L101. Deleted small amounts.

Answer: “Small amounts” was deleted (new line 118).

  1. L107. Ref to the method of fat extraction.

Answer: The reference for fat extraction was added (ref no.23, new line 129).

  1. L120. Add company name, city, and country to all devices and chemicals involved in the manuscript. Please double-check others throughout the manuscript.

Answer: This information was added for all devices and chemicals through the manuscript (highlighted in the text with red color).

  1. Results and discussion, add numbering to sub-headings.

Answer: We have numbered the sub-headings in this section.

  1. Table 1. What about the interaction between additive and heat treatment or fat type*heat treatment*additive? I think three factors should be considered in statistical analysis and their interactions.

Answer: Thank you for your suggestion. The interaction between three factors will be taken into account in our future studies in which we intended to study more concentrations of antioxidant and other heating intervals or the efficiency of burdock extract compared to other antioxidants.

  1. Table 2 and Table 3. Why did the authors separate between two fat types (lard and goose)? Otherwise, they should choose one fat type and examine it. You have three factors in your design fat type, temperature, and additive.

Answer: The two types of fats were selected as research objects based on the differences and similarities between them. They differ in the composition of polyunsaturated fatty acids (goose fat is richer compared to lard), but also they have a similar color (white-yellowish), similar consistency, and both can be used as spreads or frying fats.

  1. L84. Only after heat treatment is included in Table 4 or double-check the titles within the table.

Answer: Thank you for your observation. We have made the correspondence between the text on new lines 96-97 and the title of table 4.

  1. Discussion section the mechanism of action is lacking, or interpretation of the findings is weak. Please revise your discussion section focusing on the mechanism of action.

Answer: We have improved the discussion of the results, and also introduction and conclusion sections (highlighted in the text with red color).

  1. The conclusion section should contain a strong message or recommendation to the readers without repeating your findings and delete p values in the conclusion

Answer: We have deleted p values from conclusions and we also revised this section. Thank you for your remarks.

Reviewer 2 Report

Comments and Suggestions for Authors

1.       In this study, animal fat is used to study its antioxidant effect under high temperature conditions. The workload is heavy. However, there are several problems worth noting, which need the author to sort out the intention of this paper.

2.       In the Introduction section, the author should explain why the two kinds of animal fat were selected as research objects and conduct a literature review on them. The original article wrote about the effects of excessive frying on oil and fat, but they are not directly related to the research content and object of this paper, so the review is meaningless. It is suggested that the author re-write the background and significance directly related to the lard and goose fat, especially in Romania in traditional food.

3.       What are the main antioxidant substances in Line 114 Burdock extract? Why the addition of 100 ppm is chosen is studied in this study.

4.       For Line 116, why is the study time only 20 min?

5.       Line 171, As far as I known, the Testo 265 was produced by Germany. It may be adjusted and calculated before use. Please check.

6.       Line 180, why do we need to determine the total phenol content? Oil contains tocopherols, additives burdock contains polyphenols antioxidant substances, total phenol determination to explain what? There are both endogenous and additive substances, so the total phenol content doesn't tell us anything, right? Moreover, the basic composition of burdock extract is not explained.

7.       Line232-239 can be included in the final summary. In Line 236-239, I think the comparison of the two animal oils is a relative property, because the polyphenols in their own oils are different, and the antioxidant effects of adding burdock extract are also different.

8.       Line269, Table 1 can adjust the format like Table 2, another page. Heating temperature part In Table 1, which specific oil is measured at different temperatures? Is it lard? Is it burdock extract? The third part is easy to cause unnecessary misunderstanding.

9.       References can consider adding the latest literature.

10.    In conclusion part line 149-152 (may be not correct, not continuous), this part may be inaccurate, At the same time, the test time of only 20 minutes of burdock extract can improve the oil quality, if the high temperature is heated for one hour or longer, the result is also decreased. In addition, the full test is only heated, and the author's argument is deep-fried, which does have some expanded impact. Hope to accurately locate the research content and do not exaggerate the efficacy. It only makes sense in the context of this study.

Comments on the Quality of English Language

good.

Author Response

  1. In this study, animal fat is used to study its antioxidant effect under high temperature conditions. The workload is heavy. However, there are several problems worth noting, which need the author to sort out the intention of this paper.

Answer: Thank you for your observations. We improved the manuscript according to you suggestions.

  1. In the Introduction section, the author should explain why the two kinds of animal fat were selected as research objects and conduct a literature review on them. The original article wrote about the effects of excessive frying on oil and fat, but they are not directly related to the research content and object of this paper, so the review is meaningless. It is suggested that the author re-write the background and significance directly related to the lard and goose fat, especially in Romania in traditional food.

Answer: The two types of fats were selected as research objects based on the differences and similarities between them. They differ in the composition of polyunsaturated fatty acids (goose fat is richer compared to lard), but also they have a similar color (white-yellowish), similar consistency, and both can be used as spreads or frying fats. We also wanted to emphasize the benefits of using animal fats as a frying medium compared to vegetable oils.

We have revised the Introduction part, and also we have conducted a literature review on lard and goose fat (highlighted in the text with red color). Nevertheless, there is scarce information in the literature regarding the

stabilization of goose fat with antioxidants, and there are no studies regarding the stabilization of these fats with burdock extract.

  1. What are the main antioxidant substances in Line 114 Burdock extract? Why the addition of 100 ppm is chosen is studied in this study.

Answer: The antioxidant used in the study is burdock extract in concentration of 0.01%. We have chosen this proportion after a literature survey and we found that plant extracts such as rosemary, oregano, sage or thyme were able to inhibit lipid oxidation at 100 ppm addition in lard, for goose fat there are no reports. We have mentioned it in the introduction section of the revised manuscript.

  1. For Line 116, why is the study time only 20 min?

Answer: Thank you for your observation. The heating time was 30 min, we corrected through the manuscript, which is the cooking range for most of the foods.

  1. Line 171, As far as I known, the Testo 265 was produced by Germany. It may be adjusted and calculated before use. Please check.

Answer: We corrected the country in the case of Testo, new line 189.

  1. Line 180, why do we need to determine the total phenol content? Oil contains tocopherols, additives burdock contains polyphenols antioxidant substances, total phenol determination to explain what? There are both endogenous and additive substances, so the total phenol content doesn't tell us anything, right? Moreover, the basic composition of burdock extract is not explained.

Answer: In animal fats the total phenol content is reduced, they can come from the animal's diet and from antioxidant addition. We wanted to follow the effect of thermal treatment on polyphenols content and which of the fats is more affected. The basic composition of burdock extract was also mentioned in the introduction part of the revised manuscript.

  1. Line 232-239 can be included in the final summary. In Line 236-239, I think the comparison of the two animal oils is a relative property, because the polyphenols in their own oils are different, and the antioxidant effects of adding burdock extract are also different.

Answer: We gave up on the description in the lines 236-239 because the comparison was not adequate and lines 232-236 were included in the summary.

  1. Line 269, Table 1 can adjust the format like Table 2, another page. Heating temperature part In Table 1, which specific oil is measured at different temperatures? Is it lard? Is it burdock extract? The third part is easy to cause unnecessary misunderstanding.

Answer: The format of Table 1 has been adjusted accordingly to Table 2.

  1. References can consider adding the latest literature.

Answer: The Reference section was revised by mentioning recent literature (highlighted in the text with red color).

  1. 10. In conclusion part line 149-152 (may be not correct, not continuous), this part may be inaccurate, At the same time, the test time of only 20 minutes of burdock extract can improve the oil quality, if the high temperature is heated for one hour or longer, the result is also decreased. In addition, the full test is only heated, and the author's argument is deep-fried, which does have some expanded impact. Hope to accurately locate the research content and do not exaggerate the efficacy. It only makes sense in the context of this study.

Answer: Thank you for your observation. The conclusions section was also revised and only mentioned heat treatment instead of deep-frying in order to avoid confusion.

Round 2

Reviewer 1 Report

Comments and Suggestions for Authors

Dear authors,  

Thank you for your revisions. Still, I am not satisfied with your statistical analysis. You have three factors in your design fat type, heat treatment, and additive. However, you analyzed them separately, why?

Table 2 and 3. in the Table's first row what do you mean by type of fat? In the table heading type of fat is already mentioned. 

Comments on the Quality of English Language

Minor editing

Author Response

Dear Editor, Ms. Roxanne Deng

Thank you for the constructive comments and suggestions made by the reviewers on our manuscript The Antioxidant Effect of Burdock Extract on the Oxidative Stability of Lard and Goose Fat During Heat Treatment, authors: Pop Flavia, Thomas Dippong, manuscript number foods-2834516.

The manuscript was revised according to the reviewers’ suggestions. Please find below the list with the modifications, as requested. Changes were made by using blue color in the manuscript.

Reviewer 1

Thank you for your revisions.

  1. Still, I am not satisfied with your statistical analysis. You have three factors in your design fat type, heat treatment, and additive. However, you analyzed them separately, why?

Answer: According to your suggestions, we have also included the multifactorial analysis in the statistical interpretation of the data (Table 1) (highlighted with blue colour).

  1. Table 2 and 3. in the Table's first row what do you mean by type of fat? In the table heading type of fat is already mentioned.

Answer: In Table 2 and 3, type of fat refers to the non-additivated and additivated samples from the same fat. In order to avoid confusion, we replaced the term “type of fat” with “lard” (Table 1) and “goose fat” (Table 2) (highlighted in tables text with blue colour).

We also improved the English language through the manuscript.

Thank you for your understanding.

Best regards

Thomas Dippong

Reviewer 2 Report

Comments and Suggestions for Authors

Accept,thanks

Author Response

Thank you for the positive reviews